# Application of Pullulan and Chitosan Multilayer Coatings in Fresh Papayas

**Linyun Zhang, Chongxing Huang * and Hui Zhao** 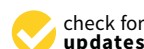

School of Light Industry & Food Engineering, Guangxi University, Nanning 530004, China;
1716303004@st.gxu.edu.cn (L.Z.); zhh@gxu.edu.cn (H.Z.)

* Correspondence: huangcx@gxu.edu.cn; Tel.: +86-771-3272256

**Abstract:** In this work, some multilayer coatings (two-layer, four-layer or six-layer) based on pullulan and chitosan for protecting papayas were prepared by the layer-by-layer technique. The papayas were coated by immersion and stored at 25 °C, 50% relative humidity or up to 14 days. Uncoated and monolayer-coated papayas were used as controls. The pullulan/chitosan coatings decreased the papaya weight loss, softening, color change ($b^*$, $\Delta E$), and pH, retarded the fall of titratable acidity and vitamin C, and maintained respiratory rate and soluble solid contents. Sensory quality evaluation demonstrated that pullulan/chitosan coatings effectively preserved papaya flavor and overall acceptance. In general, the four-layer coatings provided the best fruit preservation. In conclusion, multilayer pullulan/chitosan coatings are efficient in maintaining the post-harvest quality and prolonging the shelf life of fresh papaya.

**Keywords:** papaya; preservation; pullulan; chitosan; multilayer coating

---

## 1. Introduction

Papaya is a rich resource of many active components with nutritional value such as vitamin C, carotene, and protease [1]. The papain found in papaya can be used to treat gastritis and indigestion, improve the nutritional value and functional properties of protein, and to produce shampoo, toothpaste, and beverages. Moreover, papayas have high medicinal and industrial value [2]. However, papayas are susceptible to fungal infections, such as anthracnose and stem-end rot, which are the major causes of papaya decay [3]. Papayas easily ripen at room temperature after harvest and quickly enter the senescence stage, with browning of the skin accompanied by dark spots and soft flesh [3]. Therefore, the development of an environmental and low-cost method for papaya preservation is necessary.

Several preservation technologies such as low temperatures treatment [4], hot water treatment [5], chemical treatments [6], and edible coatings [7] have been developed for papaya preservation. Among these approaches, edible coatings have become one of the major papaya preservation methods due to their relative low cost and simple application [8]. Polysaccharide-based coatings can control the internal atmosphere of fruits and delay ripening [9], as they provide a barrier against moisture, $CO_2$ and $O_2$ [10]. Zillo et al. [3] coated papayas with a carboxymethyl cellulose solution containing *Lippia sidoides* essential oils and measured the postharvest quality attributes of coated papayas. The results showed that the coating provided a good barrier to $O_2$ and water vapor, and extended the shelf life of papaya. However, essential oils used as antimicrobial agents are expensive and volatile.

Chitosan has broad-spectrum antimicrobial properties [11] as well as permselectivity to ethylene, $CO_2$ and $O_2$. This translates into a reduction in the respiration of fruit and might reduce the loss of organic substances [12]. However, chitosan film serves as a poor barrier for water vapor, which limits the application of chitosan in fruit preservation [13]. Priyadarshi et al. [14] prepared chitosan films containing different concentrations of apricot kernel essential oil (AKEO). The results showed

that the incorporation of AKEO improved the resistance to moisture loss and water vapor barrier properties of the chitosan film. Pullulan is a nonionic polysaccharide with good adhesiveness and film formability [15]. Pullulan membranes are nearly impenetrable to gases such as $O_2$, $N_2$, $CO_2$ and aromas at low relative humidity (RH) [16]. Unfortunately, pullulan has barely any antifungal properties and is rarely used alone for fresh fruit preservation [17]. Silva et al. [16] incorporated lysozyme nanofibers into pullulan solutions to prepare a nanocomposite film. The nanofibers not only maintained the film-forming ability of pullulan but also imparted good antibacterial and mechanical properties to the nanocomposite film.

Multilayer coatings consist of different polymers that are alternately deposited on the target surface, and this preservation process is simple and inexpensive [18]. Multilayer coatings can combine the advantages of several single-layer coatings, with complete structure and stable performance [19]. Furthermore, multilayer coatings can effectively maintain the color, freshness, and firmness of fruit, thereby prolonging the postharvest life [20]. Yin et al. [21] deposited three-, five-, and seven-layer coatings based on chitosan solution containing cinnamon essential oil microcapsules and alginate solution on mango surface. The mangos were stored at 25 °C, 50% relative humidity (RH). The results indicated that multilayer coatings had dominant barrier effect to moisture and gases, maintained all the physiochemically indexes and extended the shelf life of mango effectively. Brasil et al. [22] applied a chitosan/pectin two-layer coating on the surface of fresh-cut papaya, and they added cinnamaldehyde microcapsules to the chitosan solution to enhance the antibacterial effect of the coating. The results showed that chitosan/pectin coating reduced the losses of vitamin C and total carotenoid content, and extended the shelf life of fresh-cut papaya stored at 4 °C for up to 15 d. However, available information on applications of pullulan/chitosan layer-by-layer coatings on entire fresh papaya is limited.

Hence, the aim of this study was to prepare multilayer coatings based on pullulan and chitosan and to evaluate their effectiveness for fresh papaya preservation.

## 2. Materials and Methods

### 2.1. Materials

Fresh papayas (*Carica papaya* L. cv 'Lingnanzhong') were purchased from a local fruit plantation in Nanning, Guangxi, China. The following reagents were used for this study: Chitosan (90.0% deacetylated, Sinopharm Chemical Reagent Co., Ltd., Shanghai, China); Glacial acetic acid (chemically pure, Tianjin Zhiyuan Chemical Reagent Co., Ltd., Tianjing, China); Pullulan (analytical pure, Aladdin Reagent Co., Ltd., Shanghai, China); Sodium hypochlorite solution (40% purity, Aladdin Reagent Co., Ltd., Shanghai, China); Sodium hydroxide (analytically pure, Aladdin Reagent (Shanghai) Co., Ltd., Shanghai, China); 2,6-dichlorophenolindophenol (analytical pure; Tianjin Guangfu Fine Chemical Research Institute); Oxalic acid (analytically pure, Tianjin Hengxing Chemical Preparation Co., Ltd., Tianjin, China); Kaolin (analytically pure, Tianjin Kemiou Chemical Reagent Co., Ltd., Tianjing, China). The pullulan and chitosan coatings are food grade, and can be used based on national and international regulations.

### 2.2. Preparation of Coating Solutions

The 0.5% (*w/v*) chitosan solution was prepared by dissolving 20 g of chitosan in 4 L 0.5% glacial acetic acid solution. The 0.5% (*w/v*) pullulan solution was prepared by dissolving 20 g of pullulan in 4 L of distilled water. The solutions were stirred for 3 h at room temperature and filtrated.

### 2.3. Coating of Fruits

Fresh papayas of the same size and color that had no bruises or black spots on the surface were selected, soaked in 0.3% (*v/v*) sodium hypochlorite solution for 3 min, rinsed with tap water for 2 min, and air-dried at room temperature.

The surface-disinfected papayas were randomly divided into six groups, with a control group and five test groups. Each group contained 30 papayas. Papayas were treated as described by Trevino-Garza et al. [20]. The treated papayas were dipped in 0.5% pullulan solution (*w/v*) for 5 min and the residual solution allowed to drip off for 2 min at 25 °C, which established the first layer on the surface of the fruit. The papayas were then immersed in 0.5% chitosan solution (*w/v*) for 5 min and allowed to stand at 25 °C for 2 min, which formed the second layer by hydrogen bonding between pullulan and chitosan. These steps were repeated to obtain the 4-layer and 6-layer multilayer coatings. The coated papayas were stored in a climate chamber (CLIMACELL404; Germany MMM Company, Planegg, Germany) at 25 °C and 50% RH. The physiological and nutritional attributes of papaya on days 0, 2, 4, 6, 8, 10, 12, and 14 were determined. For the experiment, 30 papayas were used per treatment during the entire storage period, and 3 fruits were used for quality mesurements per sampling day.

## 2.4. Determination of Fruit Quality

### 2.4.1. Weight loss, firmness, color and respiratory rate

A direct fruit weighing method described by Oregel-Zamudio et al. was used [23]. The papayas were weighed at each sampling time using an electronic scale (457A; Shengzhen Botoo Electronic Technology Co., Ltd., Shengzhen, China). The result was calculated as a percentage of total weight loss between the initial and final weights.

Firmness of papayas was measured as previously described by Jongsri et al. [24]. A fruit firmness meter (GY-2 type; Shanghai Hu Yueming Scientific Instrument Co., Ltd., Shanghai, China) was used to measure the firmness of each papaya by penetrating the skin with a $\Phi 3.5$ mm probe to a depth of 1 cm at three different locations on the fruit (proximal, distal, and middle).

As described by Peretto et al. [25], the color of papayas was determined using a spectrophoto-meter (CM-3600d; Japan Konica Minolta Company, Tokyo, Japan). The change in *b\** value and $\Delta E$ was defined as color alteration. The formula used to determine $\Delta E$ is as follows:

$$\Delta E = \sqrt{\left(L_i^* - L_0^*\right)^2 + \left(a_i^* - a_0^*\right)^2 + \left(b_i^* - b_0^*\right)^2}, \tag{1}$$

where $L_i^*$ represents the brightness value of papaya on day i, $L_0^*$ represents the initial brightness value of papaya, $a_i^*$ represents the red-green value of papaya on day i, $a_0^*$ represents the initial red-green value of papaya, $b_i^*$ represents the yellow-blue value of papaya on day i, and $b_0^*$ represents the initial yellow-blue value of papaya.

Respiratory rate was measured using a flow-through system according to a method previously described by Gong et al. [26]. Papayas were individually weighed and placed in a fruit respiration apparatus (JFQ-315OH; Jun-Fang-Li-Hua Technology Institute, Beijing, China) and the valve was opened to allow air flow inside. Respiration was expressed as the $CO_2$ concentration ($mg \cdot kg^{-1} \cdot h^{-1}$).

### 2.4.2. Soluble Solids Content (SSC), Titratable Acidity (TA), pH and Vitamin C (VC) Content

The SSC of papayas was determined according to the method of Khaliq et al. [27]. First, 10 g of papaya pulp and 50 mL of distilled water were mixed to extract juice using a juicer (JYL-C020E; Nine Yang Co., Ltd., Shandong, China) for 3 min. After filtration, SSC in a drop of the supernatant was measured using an Abbe refractometer (WYA-2S; Shanghai Shen Guang Instrument Co., Ltd., Shanghai, China).

The TA of papayas was measured using a method previously described by Zhao et al. [28]. First, 10 g of papaya pulp were homogenized in 50 mL of distilled water using a juicer (JYL-C020E; Nine Yang Co., Ltd., Shandong, China) for 3 min and then filtered. The supernatant was incubated in water

at 78 °C for 30 min. Next, 10 mL of mixture were titrated with 0.1 M NaOH. TA was calculated (%) with the formula below:

$$TA(\%) = \frac{\text{NaOH volume} \times 0.1 \times 0.064 \times 50 \text{ mL}}{10 \text{ g} \times 10 \text{ mL}}.$$ (2)

The pH of papayas was measured according to Temizkan et al. [29]. First, 10 g of papaya pulp was homogenized in 50 mL of distilled water using a juicer for 3 min and then filtered. The solution was collected and its pH was measured using a pH-meter (FE28; Jinan Guangyao Medical Equipment Co., Ltd., Shandong, China).

VC in papaya pulp was determined by 2,6-dichlorophenolindophenol (analytical pure; Tianjin Guangfu Fine Chemical Research Institute, Tianjin, China) titration [30]. First, 20 g of papaya pulp was mixed with 50 mL of 2% oxalic acid solution, and each sample was quickly homogenized using a juicer and filtered. Next, 20 mL of the filtrate were transferred to a 100 mL volumetric flask with 2% oxalic acid solution. Then, the solution was decolorized with 5 g of kaolin. The solution was filtered, and 10 mL of the filtrate was titrated with calibrated 2, 6-dichlorophenolindophenol until the solution turned pink and did not fade within 15 s. The VC content of papaya was calculated as follows:

$$VC = \frac{(V_i - V_0) \times T \times A}{m} \times 100,$$ (3)

where VC is the vitamin C content (mg·kg$^{-1}$) of papaya, $V_i$ is the volume (mL) of 2,6-dichlorophenolindophenol consumed in the titration of the sample, $V_0$ is the volume (mL) of 2,6-dichlorophenolindophenol consumed by blank titration, T is the titer (mg·mL$^{-1}$) of 2,6-dichlorophenolate sodium, A is the dilution factor, and m is the papaya pulp weight (g).

### 2.4.3. Sensory Quality Evaluation

The appearance, flavor, and taste of the papayas were evaluated by 10 people who were professionally trained in sensory evaluation, as described by Ma et al. [31]. The score was based on a 10-point scale where a score of 5 was considered as the limit of acceptability. Scores below 5 corresponded to off-flavor production and a poor taste of papayas.

### 2.5. Statistical Analysis

The data were analyzed using an analysis of variance (ANOVA) with the SPSS 16.0 program (SPSS Inc., Chicago, IL, USA). Statistical significance was expressed at *P* < 0.05. The figures were drawn with Origin 8.1 (OriginLab Corp., Northampton, MA, USA).

## 3. Results and Discussion

### 3.1. Weight Loss

Papaya continuously loses moisture because of transpiration; therefore, the weight loss continues to increase. As shown in Figure 1a, irrespective of the treatment, weight loss of papayas increased as the storage time was prolonged. However, weight loss of uncoated papayas was significantly higher than that of coated papayas during storage (*P* < 0.05), as previously reported by others [3]. On the 14th day, weight loss of non-treated papayas reached 28.86%. The protective effects of coatings could be attributed to the semipermeable barrier to gas exchange and water loss created on the papaya's surface [32]. Compared with papayas coated with a single-layer, papayas coated with two, four or six layers showed lower weight loss (15.53%, 10.41% and 13.85%, respectively) on the 14th day. Pullulan and chitosan can form stable coatings on the papaya surface through hydrogen bonding and the coatings become denser as the layers increase; therefore, the barrier performance is better. Unfortunately, prolonged exposure of papaya to acidic chitosan solution may damage cell walls of the

epidermis and lead to water loss of papaya, as shown in previous work with mango fruits [21]. In this work, the weight loss of the four-layer coated papayas was the smallest.

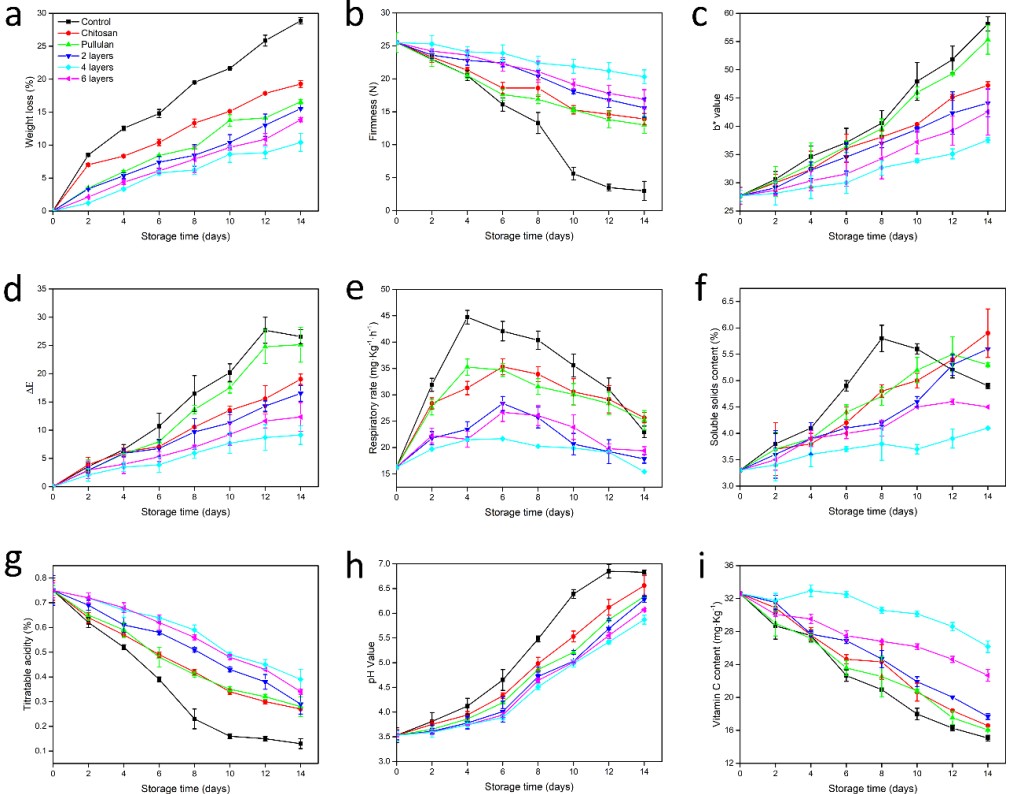

**Figure 1.** Effect of pullulan-chitosan multilayer coatings on (**a**) weight loss; (**b**) firmness; (**c**) *b\** value; (**d**) $\Delta E$; (**e**) respiratory rate; (**f**) SSC; (**g**) TA; (**h**) pH and (**i**) VC of papayas stored at 25 °C and 50% RH for 14 days. Vertical bars represent the standard errors of the means.

### 3.2. Firmness

During storage, pectin in papaya gradually decomposes by pectin degrading enzymes, which leads to a decrease in the firmness of papaya; the hydrolysis of starch in papaya can also promote fruit softening, as shown in other work with apples [33]. As shown in Figure 1b, the control samples exhibited a high rate of firmness loss compared with the coated samples ($P < 0.05$), as previously reported by others [34]. On day 14, uncoated control papayas were completely softened and could no longer be stored and transported. Papayas coated with two, four and six layers presented firmness values of 15.6, 20.2 and 16.9 N, respectively, on the 14th day, which were higher than the firmness of papayas coated with a single-layer, and this was consistent with prior research [32]. This result can be attributed to the low $O_2$ atmosphere created on the papaya's surface by the coating application, which can inhibit the activity of the enzymes involved in cell wall degradation processes and solubilization of pectins, as shown in previous work [35]. Moreover, the chitosan coating may also inhibited the activity of these enzymes [36]. As shown in Figure 1b, we also observed that the papayas coated with a six-layer coating were less firm than those coated with a four-layer coating, which may be due to the increased deposition time resulting in an adverse effect of the acidic chitosan solution to papaya epidermal cells [21]. In conclusion, the four-layer coating showed a positive effect in delaying fruit softening.

### 3.3. Color

Color is the most representative indicator of papaya maturity [37]. With prolonged storage time, chlorophyll is hydrolyzed by enzymatic action and decomposed by photooxidation, resulting in a

gradual yellowing of papaya, as shown in other work with mangoes [38]. As shown in Figure 1c, the *b**
value of the control samples increased rapidly, indicating that the papayas turned yellow quickly. On
the 14th day, *b** value of uncoated papayas reached 58.1. However, the *b** value of coated papayas
was lower and increased slower, as previously reported by others working with other fruits such
as mango [39]. Papayas coated with two, four or six layers showed *b** values of 44.1, 37.6 and 42.6,
respectively, all of which were lower than that of the single-layer coating (*P* < 0.05). The multilayer
coating created a low $O_2$ atmosphere on the surface of papaya, inhibiting the activity of respiratory
enzymes and ethylene synthetases [40]. These metabolic alterations delayed the fruit maturation
process and maintained the stability of the cell wall for longer, which translated into greater protection
to chloroplasts and consequently to chlorophylls [35]. In addition, the antimicrobial effects of the
chitosan coating may also inhibit the activity of the chlorophyllase enzyme [41]. On the 14th day, the *b**
value of four-layer coated papayas was 35.38% lower than that of uncoated papayas.

The ΔE value reflects the change in *L** and hue angle of the fruit [35]. The smaller the ΔE value,
the more stable the color change of papaya [42]. As shown in Figure 1d, if compared with that of the
control samples, the ΔE value of the coated fruits increased more slowly (*P* < 0.05), as shown in other
work with mangoes [40]. Compared to papayas coated with other layers, the ΔE of papayas treated
with a four-layer coating was minimal, and the color difference was more stable.

### 3.4. Respiratory Rate

Papaya is a climacteric fruit whose respiratory intensity changes with time and exhibits a sudden
rise followed by a drop [43]. As shown in Figure 1e, uncoated papayas quickly reached the respiratory
peak on the 4th day, and then the papaya respiratory rate rapidly decreased, as previously reported
by Li et al. [44]. However, the respiratory peaks of multilayer coated papayas were both delayed
until the 6th day, and the increase in respiration rate was suppressed compared with control fruits
(*P* < 0.05). This indicates that the multilayer coating created a modified atmosphere with high $CO_2$ and
low $O_2$ in the papaya, which reduced the respiratory rate, as shown in other work with mangoes [21].
Moreover, the antibacterial properties of the chitosan coating could have a role in the inhibition of the
activity of respiratory enzymes and delayed respiration, as shown in previous work with nectarine
fruits [45]. The respiratory rate of papaya with a four-layer coating was the most stable, better than
that of papayas with a six-layer coating, which may be due to excessive barrier properties against $O_2$
of the six layer coating, resulting in anaerobic respiration of the papayas.

### 3.5. Soluble Solids Content

As shown in Figure 1f, the SSC of uncoated papayas increased quickly during the first 8 days
of storage and then decreased rapidly, similar to that reported in previous studies with other fruits
such as mango [46]. This may be because carbohydrates in fruits are hydrolysed into sugars during
ripening, and this translates into an increase in SSC [47]. As the storage period was prolonged, the SSC
reached the maximum value. Papaya respiration continued in the later period with strong microbial
growth, resulting in a large amount of nutrient decomposition and a significant decrease in the
SSC [48]. However, the change in SSC of coated papaya was observably lower than that in the control
samples (*P* < 0.05). Compared with papayas coated with a single-layer, the SSC of papayas coated
with two-layer, four-layer, and six-layer increased slower (*P* < 0.05). The multilayer coatings formed
an effective barrier to $O_2$ in papaya, slowing down respiration and metabolic activity, and therefore
retarded fruit ripening, as shown in other work with mangoes [21]. In fact, the change in SSC in the
four-layer coated fruit group was minimal.

### 3.6. Titratable Acidity

TA content is regarded as an important indicator of respiration rate of fruits as organic acids are
substrates for the respiratory metabolim [49]. As shown in Figure 1g, the TA content of coated and
uncoated papayas decreased as the storage period was extended, as previously reported by others [49].

However, the TA of coated papayas showed a slower decrease than that of uncoated papayas ($P < 0.05$). Compared with single-layer coated papayas, at day 14, papayas coated with two-layer, four-layer, or six-layer coatings had higher TA (0.29%, 0.39% and 0.34%, respectively) ($P < 0.05$). Multilayer coatings reduced the respiratory rate of papayas, resulting in a decrease in the transformation of organic acids. Generally, four-layer coated papayas had higher TA at day 14.

### 3.7. pH

As shown in Figure 1h, pH values for all treatments increased continuously, which could be ascribed to a decline in the citric acid content of papayas [48]. However, by day 14 of storage, the pH values of coated papayas were lower than that of uncoated papayas ($P < 0.05$), which was consistent with previous results with other fruits such as mango [50]. This might be due to the gas barrier created by the coatings slowing down the metabolism of papaya and reducing the decomposition of organic acids [5]. On the 14th day, papayas coated with the four-layer coating had the lowest pH value.

### 3.8. Vitamin C

As shown in Figure 1i, the VC content of papayas declined during storage, although the coatings reduced the decrease ($P < 0.05$), as previously found by others [51]. After 14 days of storage, the VC content of control papayas decreased obviously, i.e., 15.06 mg·kg$^{-1}$, while papayas with a four-layer coating exhibited the highest VC content, i.e., 26.17 mg·kg$^{-1}$ ($P < 0.05$). The multilayer coatings created a barrier to $O_2$ and $CO_2$, delaying the oxidation of VC [16]. The antimicrobial effects of the chitosan coating may have also inhibited the activity of ascorbate oxidase [45]. In conclusion, the application of the four-layer multilayer coating effectively delayed the oxidation of VC.

### 3.9. Sensory Quality Evaluation

As shown in Figure 2, the sensory quality of the control fruits decreased quickly. The control fruits lost their commercial value on the 8th day. This is mainly attributed to their high water loss and quick softening, causing bad appearance and poor taste of the papaya [30]. The sensory quality of coated papayas declined slowly, especially in case of the four-layer coated papayas, which still were rated with 8 points on the 14th day. This might be due to the effective barrier to $O_2$ and moisture of this coating. It is worth mentioning that the sensory quality of the six-layer group decreased rapidly after eight days until the fruits lost their commercial value, which might be due to the excessive ethanol produced by anaerobic respiration of the fruit, affecting the flavor and taste of papaya.

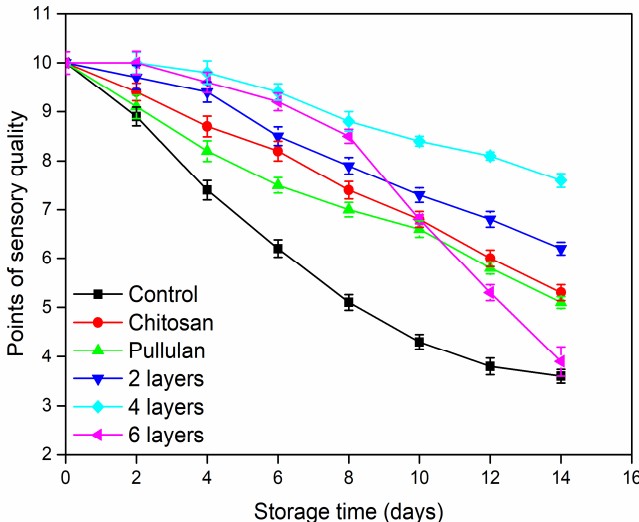

**Figure 2.** Sensory quality evaluation of papayas stored for 14 days at 25 °C, 50% RH. Scores below 5 mean that the papaya is not marketable. Vertical bars represent the standard errors of the means.

## 4. Conclusions

In summary, pullulan/chitosan multilayer coatings maintained the physiological and nutritional attributes of papayas stored at 25 °C and 50% RH, and extended the fruit shelf life if compared with uncoated and single-coated fruits. Sensory evaluation showed that the multilayer coatings maintained the flavor and commercial value of papayas for longer. This could be attributed to the ideal barrier to $O_2$ and moisture, and also to the inhibition of the activity of the respiratory enzymes. So pullulan/chitosan multilayer coatings could be applied as a new technique for fruit preservation. Among multilayer coatings, the best performance was obtained with the four-layer coating.

**Author Contributions:** Methodology, L.Z.; software, L.Z.; formal analysis, L.Z.; data curation, C.H.; writing—original draft preparation, L.Z.; writing—review and editing, H.Z.; visualization, H.Z.; supervision, C.H.

**Funding:** The authors are grateful for the financial support of the Guangxi Science and Technology Plan Project (Project No. 2018AB45007), the Opening Project of Guangxi Key Laboratory of Clean Pulp & Papermaking and Pollution Control (Project No. KF201607) and the Guangxi Natural Science Foundation Program (2018JJB120084).

**Conflicts of Interest:** The authors declare no conflict of interest. The sponsors had no role in the design, execution, interpretation, or writing of the study.

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
