# Peer review of "Application of Pullulan and Chitosan Multilayer Coatings in Fresh Papayas"

_coatings, doi:10.3390/coatings9110745_

Round 1
Reviewer 1 Report
The manuscript entitled “Application of pullulan and chitosan multilayer coatings in fresh papayas” describes the effect of edible coatings on quality of papaya during storage at 25°C for 14 days. The manuscript is well written and reports interesting results. I consider it acceptable for publication after minor revision. In fact, I would like to have more details on materials and methods used to evaluate the quality of the fruits. In particular, the authors must clarify:
If the pullulan and chitosan are food grade. Please specify in the material section if they can be used based on national and international regulation. Please specify in the introduction How the authors evaluated the formation of the film on the fruits surface Is two minutes enough to let dry the coating? How did you evaluate it? how the judges used for the sensory evaluation of the product has been trained? When using only 10 persons, they must be selected and trained for the specific product. Please give details on that.
Author Response
Date: Oct 20, 2019
Manuscript ID: coatings-614114
Title: " Application of Pullulan and Chitosan Multilayer Coatings in Fresh Papayas"
Dear Editors and Reviewers:
Thank you for your kind letter and for the reviewers’ comments concerning our manuscript entitled " Application of Pullulan and Chitosan Multilayer Coatings in Fresh Papayas ". Those comments are all valuable and very helpful for revising and improving our paper, as well as the important guiding significance to our researches. According to the reviewers’ comments, we have revised the manuscript and carefully proof-read the manuscript to minimize grammatical and bibliographical errors.
Here below are the responds to the reviewer’s comments and revised portions are marked in red in the manuscript.
Reviewers' comments:
Reviewer 1: The manuscript entitled “Application of pullulan and chitosan multilayer coatings in fresh papayas” describes the effect of edible coatings on quality of papaya during storage at 25°C for 14 days. The manuscript is well written and reports interesting results. I consider it acceptable for publication after minor revision. In fact, I would like to have more details on materials and methods used to evaluate the quality of the fruits. In particular, the authors must clarify:
If the pullulan and chitosan are food grade. Please specify in the material section if they can be used based on national and international regulation. Please specify in the introduction How the authors evaluated the formation of the film on the fruits surface. Is two minutes enough to let dry the coating? How did you evaluate it? how the judges used for the sensory evaluation of the product has been trained? When using only 10 persons, they must be selected and trained for the specific product. Please give details on that.
Response:
1). We have supplemented some instructions about chitosan and pullulan in the material section: The pullulan and chitosan are food grade, and can be used based on national and international regulation.
2). A 2×2 cm2 coating was peeled from papaya covered with four layers of multilayer coating and was subjected to cross-sectional analysis using a scanning electron microscope (SU-8020; Hitachi Hi-tech Co., Ltd., Japan) at a voltage of 10 kV after liquid nitrogen brittle fracture. As shown in Fig. 1, the coatings had smooth surface and had no holes. Moreover, the cross-section of the multilayer coating showed distinct layers with relative uniformity and tight binding, indicating that pullulan and chitosan can form a multilayer coating on the surface of papaya through hydrogen bonding.
Fig 1. SEM cross-sectional images of the four-layer coating.
3). We let the papayas stand for 2 min was to remove the residual solution, Rather than letting the coating dry.
4). The appearance, flavor and taste of the papayas were evaluated by 10 professionally trained personnel in sensory evaluation, as described by Jiachun et al. And their weights are 0.3, 0.3, 0.4, respectively. The score is based on a 10-point scale. And scores equal or higher to 5 were considered the limit of acceptability. Scores below 5 correspond to the off-flavor and poor taste of papayas.
Reviewer 2 Report
The article is intended to assess the performance of multilayer pullulan-chitosan coatings to prolong the postharvest life of fresh papayas during storage at room temperature (up to 14 d at 25ºC). After coating application, fruit weight loss, firmness, peel color, SSC, TA, vitamin C, respiration rate, and sensory quality were determined every 2 d of storage.
This kind of research is interesting since the development of novel edible compounds for postharvest preservation of fresh fruits is gaining increasing interest for the industry. Particularly, the use of chitosan and derivatives, including multilayer coatings, is especially important because of their antimicrobial properties. To my knowledge, the research in the present manuscript with papayas is new, it is generally well done, and the conclusions are sound. However, there are many aspects of the manuscript that need deep revision and considerable improvements. The Introduction and the Discussion sections are very short and weak, plenty of generalities, and with frequent improper use of literature references. Many are located in the text as if they refer to papaya studies but they refer to other fruits. Others refer to fresh-cut produce and are not applicable to entire fresh fruits. Discussion has to be deeper, explaining what is concluded in the cited prior studies, with which commodities were done and how they relate to the present data and results. In general, there are too many useless references.
Additional weak points are:
Why decay data (antifungal effect) are not provided?? Chitosan is an antimicrobial coating and decay control of fresh fruits is one of its more important functionalities. Why only data on storage at room temperature (25ºC) are provided? Commercial use of edible coatings is generally intended for fruit subjected to cold storage (postharvest life at refrigeration temperatures). The number of replicates and fruits per replicate used in the experiments has to be stated. The statistical analyses are not described. The figures presentation has to be improved. Only one legend per figure is needed. The Conclusions section has to be improved.
In my opinion, this manuscript should be published only after major revision. A comprehensive review of the manuscript, including English grammar and syntax corrections, is provided in the edited PDF file.

Author Response
Date: Oct 20, 2019
Manuscript ID: coatings-614114
Title: " Application of Pullulan and Chitosan Multilayer Coatings in Fresh Papayas"
Dear Editors and Reviewers:
Thank you for your kind letter and for the reviewers’ comments concerning our manuscript entitled " Application of Pullulan and Chitosan Multilayer Coatings in Fresh Papayas ". Those comments are all valuable and very helpful for revising and improving our paper, as well as the important guiding significance to our researches. According to the reviewers’ comments, we have revised the manuscript and carefully proof-read the manuscript to minimize grammatical and bibliographical errors.
Here below are the responds to the reviewer’s comments and revised portions are marked in red in the manuscript.
Reviewers' comments:
Reviewer 2: This kind of research is interesting since the development of novel edible compounds for postharvest preservation of fresh fruits is gaining increasing interest for the industry. Particularly, the use of chitosan and derivatives, including multilayer coatings, is especially important because of their antimicrobial properties. To my knowledge, the research in the present manuscript with papayas is new, it is generally well done, and the conclusions are sound. However, there are many aspects of the manuscript that need deep revision and considerable improvements. The Introduction and the Discussion sections are very short and weak, plenty of generalities, and with frequent improper use of literature references. Many are located in the text as if they refer to papaya studies but they refer to other fruits. Others refer to fresh-cut produce and are not applicable to entire fresh fruits. Discussion has to be deeper, explaining what is concluded in the cited prior studies, with which commodities were done and how they relate to the present data and results. In general, there are too many useless references.
Additional weak points are:
Why decay data (antifungal effect) are not provided?? Chitosan is an antimicrobial coating and decay control of fresh fruits is one of its more important functionalities. Why only data on storage at room temperature (25ºC) are provided? Commercial use of edible coatings is generally intended for fruit subjected to cold storage (postharvest life at refrigeration temperatures). The number of replicates and fruits per replicate used in the experiments has to be stated. The statistical analyses are not described. The figures presentation has to be improved. Only one legend per figure is needed. The Conclusions section has to be improved.
Response:
1). We have modified the grammar and description of the article according to the reviewer.
2). We have revised the introduction and discussion according to the suggestions of the reviewer. The revisions are marked in red.
3). References of other fruits are cited due to there are less researches on fresh papaya preservation. And the trends of physicochemical indicators of other fruits can also be used for the analysis of papaya physicochemical indicators.
4). We deleted some references.
5). Decay data (antifungal effect) are not provided due to the coated papayas stored at 25℃, 50 % RH for 14 d seldom decayed.
6). The main purpose of this experiment was to explore the protective effect of different-layer coatings on papaya. We chose 25℃ because of the limitation of the conditions at that time. As you said, we may choose 4℃ to extend the shelf life of papaya.
7). For the experiment, 30 papayas were used per treatment during the entire storage period, and 3 fruits were used for index mesurements per sampling day.
8). Only one legend per figure is used in this article.
9). The Conclusions section has been improved.
Round 2
Reviewer 2 Report
Although the manuscript has improved considerably, the authors have not followed some of the recommendations made before by reviewers and, consequently, the quality of the article is not high enough to warrant publication in Coatings.
The most important points that need substantial modifications are the following. In addition, many other editions to enhance English language and formal aspects have been made in the attached PDF file. The paper can only be published if the authors conveniently address ALL the comments in this revision.
1) The proper use of literature references is mandatory for publication. Many references are located in the text as if they refer to papaya studies when, in fact, they refer to other fruits, especially in the Discussion section. I understand that references to other fruits with postharvest behavior similar to that of papaya (climacteric fruits) can be used if no papaya studies on the particular subject that is being described are available, but then the correspondent fruit has to be identified in the text. So please revise carefully all the references used and:
1a) Please do a deeper and better literature search because there are many studies on postharvest life and coating of papaya that are not cited in this article and could be appropriate to discuss some aspects.
1b) If the reference to other fruits is maintained, then add ALWAYS in the text sentences to notice this fact. For example: “…as shown in other work with mangoes”; “…as other researchers found working with guavas”; “…as it was demonstrated on cold-stored apricots by XX et al.” ETC., ETC…
1c) Delete the references on non-climacteric fruits such as tangerines, etc. because their postharvest behavior is different than that of papaya.
1d) Delete the references on fresh-cut or minimally processed fruits because their postharvest life and physiology is not comparable to that of entire fresh fruits.
2) L70: state the name of the cultivar or variety of papaya used in this study.
3) It was recommended in the previous revision to enlarge the figures for better reading and now they have been even compressed and reduced. Please use one entire page for Fig 1 and use one unique legend (not 9 repeated identical legends).
4) The values of firmness in Fig 1b and L183 cannot be correct if the units are Newtons (N)!! They are at least 10 times larger than normal values for papaya. Please revise and change them.
5) It was recommended in the previous revision to edit the format of the references list to comply with the style of the journal Coatings, but this has not been done. So please homogenize Coatings style for all references in the list:
5a) First letter of title words NOT in capital letter.
5b)Use abbreviated journal names and in italics.
5c) Put the year in bold.
5d) Do not include the issue number after the volume number.
5e) Put in italics all the scientific names in the list.
5f) L432: do not use ref 51a and 51b.
ETC.

Author Response
Date: Nov 2, 2019
Manuscript ID: coatings-614114
Title: " Application of Pullulan and Chitosan Multilayer Coatings in Fresh Papayas"
Dear Editors and Reviewers:
Thank you for your kind letter and for the reviewers’ comments concerning our manuscript entitled " Application of Pullulan and Chitosan Multilayer Coatings in Fresh Papayas ". Those comments are all valuable and very helpful for revising and improving our paper, as well as the important guiding significance to our researches. According to the reviewers’ comments, we have revised the manuscript and carefully proof-read the manuscript to minimize grammatical and bibliographical errors.
Here below are the responds to the reviewer’s comments and revised portions are marked in red in the manuscript.
Reviewers' comments:
1) The proper use of literature references is mandatory for publication. Many references are located in the text as if they refer to papaya studies when, in fact, they refer to other fruits, especially in the Discussion section. I understand that references to other fruits with postharvest behavior similar to that of papaya (climacteric fruits) can be used if no papaya studies on the particular subject that is being described are available, but then the correspondent fruit has to be identified in the text. So please revise carefully all the references used and:
1a) Please do a deeper and better literature search because there are many studies on postharvest life and coating of papaya that are not cited in this article and could be appropriate to discuss some aspects.
1b) If the reference to other fruits is maintained, then add ALWAYS in the text sentences to notice this fact. For example: “…as shown in other work with mangoes”; “…as other researchers found working with guavas”; “…as it was demonstrated on cold-stored apricots by XX et al.” ETC., ETC…
1c) Delete the references on non-climacteric fruits such as tangerines, etc. because their postharvest behavior is different than that of papaya.
1d) Delete the references on fresh-cut or minimally processed fruits because their postharvest life and physiology is not comparable to that of entire fresh fruits.
2) L70: state the name of the cultivar or variety of papaya used in this study.
3) It was recommended in the previous revision to enlarge the figures for better reading and now they have been even compressed and reduced. Please use one entire page for Fig 1 and use one unique legend (not 9 repeated identical legends).
4) The values of firmness in Fig 1b and L183 cannot be correct if the units are Newtons (N)!! They are at least 10 times larger than normal values for papaya. Please revise and change them.
5) It was recommended in the previous revision to edit the format of the references list to comply with the style of the journal Coatings, but this has not been done. So please homogenize Coatings style for all references in the list:
5a) First letter of title words NOT in capital letter.
5b)Use abbreviated journal names and in italics.
5c) Put the year in bold.
5d) Do not include the issue number after the volume number.
5e) Put in italics all the scientific names in the list.
5f) L432: do not use ref 51a and 51b.ETC.
Response:
1) We revised carefully all the references used, and the modified parts are distinguished by red.
1a) Many studies on postharvest life and coating of papaya that are cited.
1b) Some references to other fruits is maintained, and we added ALWAYS in the text sentences to notice this fact.
1c) We deleted the references on non-climacteric fruits such as tangerines, etc. because their postharvest behavior is different than that of papaya.
1d) We deleted the references on fresh-cut or minimally processed fruits because their postharvest life and physiology is not comparable to that of entire fresh fruits.
2) The variety of papaya is Carica papaya L..
3) We have provided the original files of figure 1 in this article. These pictures can be enlarged or reduced according to your requirements.
4) We have reviewed the experimental datas and found that the values of firmness have been magnified ten times. The values of firmness have been modified.
5) We modified the format of the references
6) We revised the article according to the PDF file provided by the reviewer, and the modified parts are distinguished by red.

Round 3
Reviewer 2 Report
The authors have improved the manuscript, but they still have not followed all the recommendations made by the reviewers, so the article still needs further editions.
The following editions have not been followed:
1) L72: state the name of the cultivar or variety of papaya used in this study. Carica papaya is the scientific name of the species, not the cultivar. Examples of well-known papaya cultivars or varieties are: Solo, Eksotika, Sunrise, Wainamalo, Kapoho, Sunnyback, Tainung, Betty, Maradol, etc., etc….
As any scientific name, the species name (Carica papaya) is in italics and the author name or abbreviation (L.) is not in italics.
2) It was recommended in the previous revision to use one entire page for Fig 1 and use one unique legend (not 9 repeated identical legends). Now they have put Fig 1 in several different pages (2 subfigures per page) and still use 9 legends (one for each subfigure).
3) It was recommended in the previous revision to edit the format of the references list to comply with the style of the journal Coatings, but this has not been completely done for:
3a) First letter of title words NOT in capital letter.
3b) Use abbreviated journal names and in italics.
3c) Do not include the issue number after the volume number.
3d) Put in italics all the scientific names in the list.
Please refer to the edited attached PDF file.

Author Response
Date: Nov 5, 2019
Manuscript ID: coatings-614114
Title: " Application of Pullulan and Chitosan Multilayer Coatings in Fresh Papayas"
Dear Editors and Reviewers:
Thank you for your kind letter and for the reviewers’ comments concerning our manuscript entitled " Application of Pullulan and Chitosan Multilayer Coatings in Fresh Papayas ". Those comments are all valuable and very helpful for revising and improving our paper, as well as the important guiding significance to our researches. According to the reviewers’ comments, we have revised the manuscript and carefully proof-read the manuscript to minimize grammatical and bibliographical errors.
Here below are the responds to the reviewer’s comments and revised portions are marked in red in the manuscript.
Reviewers' comments:
1) L72: state the name of the cultivar or variety of papaya used in this study. Carica papaya is the scientific name of the species, not the cultivar. Examples of well-known papaya cultivars or varieties are: Solo, Eksotika, Sunrise, Wainamalo, Kapoho, Sunnyback, Tainung, Betty, Maradol, etc., etc….
As any scientific name, the species name (Carica papaya) is in italics and the author name or abbreviation (L.) is not in italics.
2) It was recommended in the previous revision to use one entire page for Fig 1 and use one unique legend (not 9 repeated identical legends). Now they have put Fig 1 in several different pages (2 subfigures per page) and still use 9 legends (one for each subfigure).
3) It was recommended in the previous revision to edit the format of the references list to comply with the style of the journal Coatings, but this has not been completely done for:
3a) First letter of title words NOT in capital letter.
3b) Use abbreviated journal names and in italics.
3c) Do not include the issue number after the volume number.
3d) Put in italics all the scientific names in the list.
Please refer to the edited attached PDF file.
Response:
1) We added the name of the cultivar of papaya used in this study (Carica papaya L. cv ‘Lingnanzhong’)
2) We use one unique legend now, and we combined the pictures without compressing their quality. Moreover, you can make your own combinations of the pictures at the end of the text as your requirements.
3) We revised the reference according to the PDF file.